# A Comprehensive Review of Advanced Diagnostic Techniques for Endometriosis: New Approaches to Improving Women’s Well-Being

**DOI:** 10.3390/medicina60111866

**Published:** 2024-11-14

**Authors:** Greta Kaspute, Egle Bareikiene, Urte Prentice, Ilona Uzieliene, Diana Ramasauskaite, Tatjana Ivaskiene

**Affiliations:** 1Department of Nanotechnology, State Research Institute Center for Physical Sciences and Technology (FTMC), Sauletekio Ave. 3, LT-10257 Vilnius, Lithuania; gretakaspute@gmail.com (G.K.); urte.prentice@gmail.com (U.P.); 2State Research Institute Centre for Innovative Medicine, Santariskiu St. 5, LT-08410 Vilnius, Lithuania; ilona.uzieliene@imcentras.lt (I.U.); tatjana.ivaskiene@imcentras.lt (T.I.); 3Department of Physical Chemistry, Faculty of Chemistry and Geosciences, Institute of Chemistry, Vilnius University, Naugarduko St. 24, LT-03225 Vilnius, Lithuania; 4Faculty of Medicine, Vilnius University, M. K. Ciurlionio St. 21/27, LT-03101 Vilnius, Lithuania; diana.ramasauskaite@santa.lt

**Keywords:** endometriosis, diagnostics, nanotechnology, artificial intelligence, biomarkers

## Abstract

According to the World Health Organization (WHO), endometriosis affects roughly 10% (190 million) of reproductive-age women and girls in the world (2023). The diagnostic challenge in endometriosis lies in the limited value of clinical tools, making it crucial to address diagnostic complexities in patients with suggestive symptoms and inconclusive clinical or imaging findings. Saliva micro ribonucleic acid (miRNA) signature, nanotechnologies, and artificial intelligence (AI) have opened up new perspectives on endometriosis diagnosis. The aim of this article is to review innovations at the intersection of new technology and AI when diagnosing endometriosis. Aberrant epigenetic regulation, such as DNA methylation in endometriotic cells (ECs), is associated with the pathogenesis and development of endometriosis. By leveraging nano-sized sensors, biomarkers specific to endometriosis can be detected with high sensitivity and specificity. A chemotherapeutic agent with an LDL-like nano-emulsion targets rapidly dividing cells in patients with endometriosis. The developed sensor demonstrated effective carbohydrate antigen 19-9 detection within the normal physiological range. Researchers have developed magnetic iron oxide nanoparticles composed of iron oxide. As novel methods continue to emerge at the forefront of endometriosis diagnostic research, it becomes imperative to explore the impact of nanotechnology and AI on the development of innovative diagnostic solutions.

## 1. Introduction

Endometriosis affects a significant proportion of reproductive-age women globally, with variations in prevalence across regions linked to healthcare accessibility [1]. Developed nations, benefitting from better healthcare access, tend to exhibit higher numbers of diagnosed cases, emphasizing the necessity for targeted studies and screening tools in less privileged regions, like Oceania [2,3]. Endometriosis is a chronic inflammatory condition marked by the presence of endometrial tissue outside of the uterus [4]. This is associated with diverse risk factors, such as familial predisposition [5], early onset of menstruation [6], genetic markers, like the LIN28B polymorphism [7], short menstrual cycles, extended and heavy periods [8], infertility [9], increasing age, smoking, alcohol use, sexual activity during menstruation, and low body weight [10,11,12].

Endometriosis diagnostics face limitations in terms of clinical examinations, questionnaires, and imaging tests, making it crucial to address diagnostic complexities in patients with suggestive symptoms and inconclusive clinical or imaging findings [9,10]. Treatment for endometriosis aims to alleviate pain, restore fertility, and regulate menstrual cycles through various approaches, such as conservative monitoring, medication, and surgical interventions [10]. Medical management focuses on reducing inflammation, suppressing ovarian cycles, and inhibiting estrogen’s effects, while surgical interventions may involve lesion removal or complete excision of pelvic organs, with ongoing debates about their long-term efficacy and potential impact on disease recurrence or progression. Unfortunately, neither medical nor surgical options universally offer long-term relief for all patients [13,14]. Therefore, the need for early diagnostics, including, especially, non-invasive options, is crucial. Nanotechnology and artificial intelligence (AI) can be useful tools to cope with diagnostic challenges without surgical intervention. For example, these might include nano-sized sensors for biomarkers specific to endometriosis detection [13] or an AI-based tool to help clinicians with diagnostics [14]. The aim of this review is to describe advances in endometriosis diagnostic techniques that impact women’s well-being.

## 2. Endometriosis: Pathogenesis and Symptom Management

According to World Health Organization (WHO) data, endometriosis affects roughly 10% (190 million) of reproductive-age women and girls in the world (2023) [1]. In 2019, the highest numbers of endometriosis cases (per 100,000 population) at the regional level were in Oceania (67.94), Eastern Europe (65.48), and Central Asia (60.87). The lowest patient numbers were in North America (31.23), East Asia (36.41), and Central Europe (38.03) [2]. These numbers are diverse because of the inconsistencies in the level of healthcare among countries. In developed countries, women may have better access to healthcare due to socioeconomic status [15], and they are more likely to be diagnosed with endometriosis [2,3].

Endometriosis is a chronic inflammatory disease defined as the expansion of endometrial stroma and functioning glands outside of the uterus cavity [16]. Prior studies have identified a variety of endometriosis risk factors, including family history when first-degree relatives are diagnosed with endometriosis (recurrence risk of 5–7%) [9]. Some studies suggest that early menarche could also be a risk factor [16]. Early estrogen stimulus may be a predisposing factor for pathologies, such as endometriosis, asthma, insulin resistance, and short stature. Sixteen papers were analyzed that showed an association between early menarche (<12 years) and a significant pooled risk of endometriosis, with high heterogeneity (OR = 1.34, 95% CI 1.16–1.54, I^2^ = 72.0%). It occurred in low-income countries. Another study showed that there is a probability of 55% that a woman with endometriosis had earlier menarche than one without endometriosis if both were randomly chosen from a population [17,18]. A single nucleotide polymorphism on chromosome 6, LIN28B, has been found to be associated with earlier menarche [5]. Other risk factors for endometriosis include short monthly cycles (less than 27 days), heavy menstrual periods that last more than 7 days [8], increasing age, smoking, alcohol use, intercourse during menstruation, and low body weight [10,11,12]. Endometriosis can be the cause of infertility [9].

Aberrant epigenetic regulation, such as DNA methylation in endometriotic cells (ECs), is associated with the pathogenesis and development of endometriosis. However, the relationship between estrogen and endometriosis is complex, as the absence of estrogen does not always mean the absence of endometriosis [19,20]. Overexpression of estrogen receptor-β (Erβ) in stromal ECs inhibits tumor necrosis factor α (TNFα)-mediated apoptosis, acts as a suppressor of (estrogen receptor-α) Erα, induces interleukin-1, and co-stimulates Ras-related estrogen-regulated growth inhibitor and serum and glucocorticoid-regulated kinase 1 as key Erβ targets with co-stimulating prostaglandin E2 under the action of estradiol. De novo increase of E2 in endometriosis lesions affects the ratio of Erα and Erβ, thus impacting inflammation and the expression of some target genes, such as Growth regulation by estrogen in breast cancer 1 (Greb-1) protein and multifunctional transcription factor oncogene (c-Myc), that result in endometriosis’ progression [20,21]. Increased oxidation of lipoproteins has been associated with the pathogenesis of endometriosis, where reactive oxygen species (ROS) cause lipid peroxidation that leads to DNA damage in endometrial cells [22]. The presence of water and electrolytes in the increased peritoneal fluid volume in patients with endometriosis harbors the source of ROS. These patients also have iron overload in their peritoneal cavities from the breakdown of hemoglobin, which in turn causes redox reactions [23]. The release of the pro-inflammatory products and the oxidative stress signals generated from the ROS cause inflammation, which leads to the recruitment of lymphocytes and activated macrophages producing cytokines that induce the oxidization of enzymes and promote endothelial growth [24]. This inflammatory response may also cause defective “immune surveillance” that prevents the elimination of menstrual debris and promotes the implantation and growth of endometrial cells in the ectopic sites [25]. The survival and resistance to immune-cell-mediated responses of ECs are ensured by masking these ectopic cells to the immune system, where, for example, ectopic ECs modulate the expression of human leukocyte antigen (HLA) class I molecules [26]. In addition to the decreased scavenger activity, the endometrium in patients with endometriosis expresses higher levels of anti-apoptotic factors. The inhibition of the apoptosis of endometrial cells may also be mediated by the transcriptional activation of genes that normally promote inflammation, angiogenesis, and cell proliferation [27].

## 3. Current Diagnostics Situation

Previous studies have highlighted the limited diagnostic value of clinical examinations and questionnaires as well as the low accuracy of imaging examinations for the detection of endometriosis, especially the superficial peritoneal phenotype (SE). The SE is the most common form of endometriosis, comprising approximately 80% of all diagnosed endometriosis. It forms as a shallow lesion along the peritoneum (the membrane that lines the abdominal cavity) that is not visible when performing TVUS [28]. Therefore, the challenge is not only to confirm advanced-stage disease but also to overcome the diagnostic complexities of patients experiencing symptoms suggestive of endometriosis but with non-contributive clinical and/or imaging examinations [9,10].

The clinical examination of the patient, including, in particular, patients’ complaints, serves as the primary basis of endometriosis diagnostics. Gynecologists in their daily work most often rely on clinical examination and TVUS in order to find endometriomas or a changed location of the ovaries that can lead to the mindset that formed adhesions may be due to endometriosis. Magnetic resonance is not used that often due to its costs and its varying availability in different countries. Diagnostic laparoscopy remains a commonly used method to confirm endometriosis, along with visual assessment and histological examination (Table 1).

## 4. Novel Methods to Diagnose Endometriosis

In recent years, as researchers and medical practitioners strive to overcome the challenges associated with timely and accurate diagnosis, interdisciplinary collaborations have paved the way for innovations at the intersection of nanotechnology and AI [34].

Nanotechnology offers unparalleled possibilities for early detection of endometriosis through non-invasive or minimally invasive techniques. By leveraging nano-sized sensors, biomarkers specific to endometriosis can be detected with high sensitivity and specificity [35]. This provides clinicians with unprecedented insights into disease progression while minimizing patient discomfort. Simultaneously, AI has emerged as a transformative force in healthcare due to its ability to analyze vast amounts of data with speed and precision. Machine learning (ML) algorithms empower AI systems to recognize intricate patterns within complex datasets derived from various sources, such as genetic profiles, imaging studies, clinical records, and patient-reported outcomes [36]. By integrating these diverse data streams through sophisticated AI models, clinicians can harness actionable information for precise diagnosis and personalized treatment plans. As novel methods continue to emerge at the forefront of endometriosis diagnostic research, it becomes imperative to explore the impact of nanotechnology and AI on the development of innovative diagnostic solutions.

## 5. Biomarkers

The identification of reliable biomarkers for non-invasive detection has been the subject of extensive research. Numerous studies have explored various biomarkers, leading to significant advancements; however, conclusive results have remained elusive. Rather than focusing on individual biomolecules, most research efforts have emphasized a panel of biomarkers as potential diagnostic tools. Despite these efforts, no singular biomarker or combination of biomarkers has demonstrated sufficient specificity and sensitivity for accurate endometriosis diagnosis [37]. Notably, cancer antigen (CA)-125, CA-199, interleukin (IL)-6, and urocortin have been extensively studied among numerous biomarkers.

CA-125 is a protein that is often elevated in the blood of individuals with endometriosis and other conditions, including ovarian cancer and pelvic inflammatory disease. While it is not specific to endometriosis, elevated levels can indicate the presence of endometrial-like tissue outside of the uterus.CA 19-9 is similar to CA-125, and it is another tumor marker that can be elevated in various gastrointestinal and some gynecological conditions. Its role in endometriosis is still under investigation.IL-6 is a cytokine involved in inflammation and immune responses. Increased levels of IL-6 have been observed in patients with endometriosis. IL-6 can contribute to the inflammatory milieu characteristic of endometriosis and reflect disease severity.Urocortin is a neuropeptide that is associated with the stress response, and it has been implicated in various reproductive processes. Some studies have shown that urocortin levels may be altered in individuals with endometriosis. The exact mechanisms and the role of urocortin in endometriosis are not entirely understood.

However, despite their prominence in investigations, no biomarker has yet been validated as clinically reliable for diagnostic applications [38], especially non-invasive applications.

Although previous reviews and international guidelines do not currently recommend the use of diagnostic biomarkers for endometriosis due to their previously reported low accuracy, the preliminary identification of a saliva micro ribonucleic acid (miRNA) signature has opened up new perspectives [39]. This signature, composed of 109 miRNAs, was developed through a single-center, prospective study of women with symptoms suggestive of endometriosis. The miRNA signature was developed using a combination of two technologies—next-generation sequencing and AI—and was found to have a sensitivity, specificity, and area under the receiver operating characteristic curve of 96.7%, 100%, and 0.98, respectively [39].

## 6. Nanotechnology’s Impact on Endometriosis Diagnostics Development

Nanotechnology presents a versatile approach to address limitations in current diagnostic methods for endometriosis, such as invasiveness, cost-effectiveness, and accuracy. By employing nanoparticle-based agents, researchers explore the non-invasive detection of endometriosis-associated biomarkers, thus bypassing the need for surgical intervention. Furthermore, coating nanoparticles with specific ligands enhances imaging techniques, thus facilitating precise visualization and localization of the disease within reproductive tissues. Nanoparticles hold the potential to revolutionize diagnostics by offering accurate and non-invasive tools for endometriosis detection and treatment [40,41].

Researchers explored the potential of combining a chemotherapeutic agent with an LDL-like nano-emulsion (LDE) to target rapidly dividing cells in diverse cancer and inflammatory conditions. The study enrolled 14 patients with intestinal or non-intestinal endometriosis, analyzing their lipid profiles before surgery and administering labeled LDE emulsions for radioactivity assessment in tissue samples. The results indicated higher plasma LDL levels but reduced LDE uptake in intestinal endometriosis, suggesting decreased cell division and heightened fibrosis. Notably, LDE uptake was highest in the topical endometrium and healthy peritoneum, with minimal uptake in endometriotic lesions (ELs). These findings underscore the potential of nanotechnology for managing deep endometriosis with surgical procedures and hormonal blockade, offering advantages like lower complication rates and the absence of systemic side effects compared to conventional treatments [32].

As mentioned earlier, many studies have focused on using biomarkers associated with endometriosis [38,42,43,44,45,46,47,48,49,50,51]. Label-free electrochemical immune-sensor was reported to be used in endometriosis diagnosis applications [42]. The developed sensor demonstrated effective carbohydrate antigen 19-9 detection within the normal physiological range. The sensitivity of the sensor suggested its potential application in early-stage endometriosis diagnostics, disease monitoring, and therapy optimization. To establish its clinical usability, researchers tested the sensor using real samples and compared its performance to that of the enzyme-linked immunosorbent assay, where satisfactory results were obtained (Table 2) [42]. Nanotechnology-based biosensors offer enhanced sensitivity and accuracy in detecting these biomarkers from various patient samples, like blood, urine, or feces [41].

A novel nanoplatform for endometriosis delineation and ablation integrates real-time near-infrared fluorescence imaging with photothermal therapy, utilizing silicon naphthalocyanine dye encapsulated within polymeric nanoparticles. In vitro and in vivo studies demonstrated successful activation of fluorescence upon internalization by endometriosis cells, enabling precise demarcation and targeted ablation of endometriotic tissues through near-infrared (NIR) light exposure, showcasing its potential for intraoperative identification and treatment [57].

The connection between the microbiome and endometriosis has been briefly investigated. The latest reports from Japan from Nagoya University show interesting relationships between fusobacterium and endometriosis. The report concludes that compared with the control group and 10% positive answers, 64% of patients with endometriosis are positive for fusobacterium in the endometrium, suggesting that this bacterial infection induces a phenotypic transition of endometrial cells and that transformed endometrial cells that reach the abdominal cavity or ovarian surface during retrograde menstruation develop endometrial lesions [59]. Fusobacterium is a part of the normal gut microbiome, but when it migrates to the genital tract, it can cause structural changes in fibroblasts of endometrium that are specific to endometriosis. In a mouse model of endometriosis, inoculation of fusobacterium increased the numbers and weights of endometriotic lesions, whereas antibiotic treatment with metronidazole and chloramphenicol reduces the lesions [59,60].

In addition to biomarker detection, imaging techniques play a crucial role in diagnosing endometriosis by visualizing lesions. Researchers [58] developed magnetic iron oxide nanoparticles composed of iron oxide and coupled them with a kinase insert domain receptor [61] These magnetic iron oxide nanoparticles were designed to specifically accumulate in ELs by targeting the vascular endothelial growth factor receptor 2 [58]. In their study, a system of kinase inserts domain receptors and magnetic iron oxide nanoparticles demonstrated the ability to selectively heat up to temperatures exceeding 50 °C when exposed to an alternating magnetic field, leading to cell death within the lesions. Additionally, these nano-vectors exhibited promising potential as contrast agents for magnetic resonance imaging in the diagnosis of endometriosis prior to applying the alternating magnetic field treatment [58].

While nanotechnology offers the advantage of non-invasiveness compared to surgical methods like laparoscopy, the long-term consequences of nanoparticle retention need to be carefully considered. Nevertheless, the use of nanotechnology-based platforms holds promise for improving the detection and diagnosis of endometriosis in a more accurate, sensitive, and accessible manner [41].

## 7. AI’s Role in Endometriosis Diagnostics

In the realm of diagnosing endometriosis, the emergence of smart healthcare has paved the way for a wealth of opportunities to leverage AI to collect and analyze vast amounts of data. With the continuous production of big healthcare data from an array of sensors, devices, and communication technologies, there arises a need for automated information fusion [62]. This process entails integrating multiple sources of information, thus leading to more reliable, effective, and precise insights that facilitate optimal decision making. A comprehensive and coherent review [14] described and distinguished three main applications of AI in terms of endometriosis: prediction, diagnostics, and improving research and monitoring. However, because this review is focused on diagnostics, the studies reviewed below will be related to AI in the diagnosis of endometriosis in patients.

A highly sensitive deep ML diagnostic model based on hub genes was applied [63]. Because the identification of a single gene or pathway underlying complex traits is difficult, researchers used the integration of gene expression methods to form a specific network. As a result, weighted gene co-expression network analysis was used to build gene modules and associate them with integrated clinical traits [63]. After pre-processing and normalization of the datasets, the neural networks were applied in that study. Other researchers conducted a prospective ENDO-miRNA study [64] employing ML methodologies to examine the human miRNome. In that paper, the authors present the initial blood-based diagnostic signature derived from a combination of two robust technologies that merge the intrinsic qualities of miRNAs with AI’s modeling capabilities [64]. The study comprised two main components: (i) biomarker discovery involving genome-wide miRNA expression profiling through small RNA sequencing utilizing next-generation sequencing, and (ii) the development of an ML algorithm to construct an accurate miRNA diagnostic signature based on expression and precision profiling [64]. Another example of using ML is a study of self-reported-symptom-based endometriosis prediction [35]. The ML techniques, such as Decision Tree (DT), Random Forest (RF), Gradient Boosting Classifier, and Adaptive Boosting, were used to train prediction models using questionnaire data collected from two groups: women diagnosed with endometriosis and women without a diagnosis [36]. The goal was to integrate this model into a website, thereby allowing individuals to freely use it as a self-diagnostic tool, and to expedite the time to diagnosis by identifying women with a high probability of having endometriosis and referring them for further examination [36]. A novel ensemble ML classifier named GenomeForest, based on chromosomal partitioning, was developed and applied to classify patients with endometriosis versus control patients, utilizing 38 RNA-seq and 80 enrichment-based DNA-methylation datasets. The classifier successfully identified candidate biomarker genes with exceptional score metrics, and the same research group employed various ML techniques, such as DT, partial least squares discriminant analysis, support vector machine, and RF models, to evaluate their performance in classifying endometriosis cases versus controls using both transcriptomics and methylomics data [65].

The support vector algorithm implemented in the R programming language library facilitated the comparison of significantly expressed metabolites between ovarian endometriosis samples and those without endometriosis. Through a comparative untargeted lipidomic analysis of human endometrial fluid, employing ultrahigh performance liquid chromatography coupled with mass spectrometry, a predictive model utilizing differentially expressed metabolites accurately classified 86% of the samples, revealing distinctive lipidomic profiles associated with ovarian endometriosis and suggesting the potential of endometrial fluid analysis as a minimally invasive diagnostic approach for endometriosis [66]. In another study utilizing a dataset of 627,566 clinically collected instances from endometriosis cases (0.82%) and control subjects (99.18%), researchers developed and evaluated predictive models aiming to create an ML platform by incorporating algorithms like logistic regression, DT, RF, AdaBoost, and XGBoost, alongside Shapley Additive Explanation values for feature importance quantification. The XGBoost model exhibited superior performance during model selection, highlighting the potential of ML in enhancing endometriosis diagnosis, although further research is needed to enhance predictive model efficacy in this field [67].

Scientists employ AI technologies like Logistic Regression, DT, RF, and other methods for diagnostic purposes for several compelling reasons. Firstly, they are extensively used for binary classification tasks because they yield interpretable outcomes by estimating the probabilities of different results [68]. This allows researchers to understand the influence of predictor variables on the likelihood of a specific diagnosis. Secondly, they offer, as a DT method, an intuitive and easily visualized approach to understanding the decision-making process within diagnostic models. Moreover, RF serves as an ensemble learning method that combines multiple decision trees to enhance predictive accuracy. The RFs provide insightful information about feature importance, thus facilitating the identification of key variables crucial for accurate diagnosis [69]. The robustness of these methods across various domains and datasets is another advantage. They can handle different types of data while displaying a relatively high tolerance towards outliers or noise present in the dataset [70]. Scalability is a noteworthy feature of the mentioned models. These and other AI techniques efficiently handle large-scale datasets without compromising computation time or performance. By utilizing these methods, scientists can develop diagnostic models that are interpretable, efficient, robust, and capable of accurately classifying patients based on symptoms or other relevant factors related to specific diseases or conditions, such as endometriosis. However, accurate diagnosis may also face difficulties due to overlapping data in databases and information fusion. The newest AI technologies can not only help but also confuse researchers if they do not have the skills necessary to work with large amounts of data from different sources [71].

## 8. Infertility

Endometriosis affects fertility. It is common when a patient comes to a gynecological appointment when having problems conceiving and endometriosis is found. In most cases, it is recommended to perform a diagnostic laparoscopy and, if needed, to perform chromotubation or remove or vaporize the masses of endometriosis to improve fertility in women who have mild or minimal endometriosis [57]. Laparoscopy can be offered as a treatment option for endometriosis-associated infertility in stage I–II endometriosis as it improves the rate of ongoing pregnancy according to the Revised American Society for Reproductive Medicine classification. The fear of ovarian failure following cystectomy has driven clinicians to perform ablative techniques, such as CO_2_ fiber laser vaporization. In this surgical approach, the endometrium is not removed but rather ablated with energies with little thermal spread [72]. The effect of cystectomy on ovarian reserve markers in terms of the antimullerian hormone (AMH) has been analyzed by Younis et al. [73]. The AMH reduction was significantly greater after bilateral cystectomy compared to unilateral cystectomy, with decreases of 53.9% vs. 38.4% in the short term and 43.4% vs. 26.9% in the intermediate term. CO2 fiber laser vaporization, showing no change in AMH levels or ovarian volume compared to cystectomy in unilateral endometrioma, appears promising for preserving ovarian function and improving fertility by stimulating intracellular signaling and promoting neo-angiogenesis [73]. The American College of Obstetricians and Gynecologists (ACOG) does not recommend using oral contraceptive pills or gonadotropin-releasing hormone (GnRH) agonists to treat endometriosis-related infertility because they prevent ovulation and delay pregnancy [19].

## 9. Stem Cells and Extracellular Vesicles

Current therapeutic approaches often focus on managing symptoms with drugs or surgery rather than addressing the underlying pathology of endometriosis. Therefore, alternative treatment methods have been of great interest during the last couple of decades. One of the research objects are stem cells, which hold promise as a novel treatment strategy for endometriosis [74,75].

Preclinical studies have demonstrated the ability of mesenchymal stromal cells (MSCs) to attenuate inflammation, promote uterus tissue repair, and modulate aberrant immune responses, all of which are implicated in the pathogenesis of endometriosis [76]. It was shown that adipose-derived MSCs efficiently mitigated endometriosis associated with chronic inflammatory reactions via reduction of CD68-positive macrophages and the expression of the proinflammatory cytokines [77]. MSC-based therapies offer several potential mechanisms of action. Firstly, MSCs have the ability to home to sites of inflammation and injury within the pelvic cavity, where they may exert anti-inflammatory effects and facilitate the resolution of ELs [78]. Secondly, MSCs can differentiate into various cell types, including endometrial-like cells, thereby replenishing damaged or dysfunctional endometrial tissue [79]. The MSCs secrete a myriad of bioactive factors, such as growth factors, cytokines, and extracellular vesicles, which can modulate local immune responses, promote angiogenesis, and stimulate tissue regeneration. It was shown that menstrual-blood-derived MSC exosomes significantly inhibit inflammation and suppress proliferation, migration, and angiogenesis in endometriosis cells [80]. Additionally, it has been demonstrated that MSCs from ELs possess more immunosuppressive properties than MSCs from the ectopic endometrium by promoting M2 macrophage growth via paracrine factors [81].

## 10. Conclusions and Future Perspectives

Follow-up and psychological support should be considered for women with confirmed endometriosis, particularly deep and ovarian endometriosis. There is currently no evidence of the benefit of regular long-term monitoring for the early detection of recurrence, complications, or malignancy. While the Ca-125, CA-199, interleukin-6, and urocortin biomarkers show promise, it is important to note that no single marker is definitively diagnostic for endometriosis. Also, the same is true for microbiota and nanotechnologies. The preliminary identification of a saliva micro ribonucleic acid signature has opened up new perspectives on diagnosing endometriosis without radiological technologies and surgical interventions. The convergence of nanotechnology and AI holds immense potential for revolutionizing endometriosis diagnostics on multiple fronts by working in AI-driven analysis algorithms that can decode subtle patterns present in medical images or biomolecular data that may elude human observation alone. Through nanomaterial-based imaging agents or biosensors, targeted delivery systems can be developed to detect small quantities of disease-related biomarkers in bodily fluids. Specialists still cannot rely on it unconditionally because of the lack of sensitivity of all markers and novel diagnostic tools. The application of stem cells, particularly MSCs, represents a promising avenue for the development of novel therapeutic approaches for endometriosis. By harnessing the regenerative and immunomodulatory properties, researchers aim to address the underlying pathophysiology of endometriosis and provide patients with more effective and durable treatment options in the future. Despite these novel techniques, the identification and diagnosis of endometriosis clinically usually rely on a combination of clinical symptoms, imaging studies, and, in many cases, surgical evaluation. While widely used diagnostic methods have a long-standing history of effectiveness, there is a pressing need for novel and non-invasive techniques to emerge and compete with them to enhance patient care and comfort in the diagnostic process. Continuation of the science is needed to purify and specify non-invasive methods for diagnosing endometriosis. If non-invasive diagnostic methods became ones that specialists could rely on, then fewer diagnostic laparoscopies would be conducted, resulting in fewer post-operative complications and lower costs.

## Figures and Tables

**Table 1 medicina-60-01866-t001:** Currently used tools for endometriosis diagnostics.

Diagnostic Tool	Description
Clinical examination	Objective examination includes painful, “pulled” rough sacro-utherine ligaments, uterus in retroflexion position, enlarged (size 6–10 weeks) uterus, foci of endometriosis in the cervix, vagina, vulva, or postoperative scar, enlarged ovaries, and palpable mass in the small pelvis (endometrioma) [19,29,30].
Transvaginal ultrasound (TVUS)	This is the primary diagnostic tool used to identify endometriomas, characterized by liquid-filled formations with thicker walls, heterogeneous or solid echostructures, and brighter peripheral shadows, often assessed alongside dopplerometry for resistive index values, typically conducted during the late follicular phase, with rectal ultrasound examination considered for suspected endometriosis in the cul-de-sac or rectovaginal area [31,32].
Magnetic resonance imaging (MRI)	MRI is most useful for ultrasonographically indeterminate pelvic masses. MRI is also superior to ultrasound in diagnosing rectosigmoid lesions and endometriosis of the bladder [33].
Diagnostic laparoscopy	Laparoscopy is no longer the diagnostic gold standard, and it is now only recommended in patients with negative imaging results and/or where empirical treatment was unsuccessful or inappropriate [29].

**Table 2 medicina-60-01866-t002:** Endometriosis diagnosis methods.

Operating Principle	Material	Detection	Ref.
Cellular internalization	Lipid core nanoparticles contained within labeled low-density lipoprotein (LDL)	Chemotherapeutic agent carried in an LDE (lipid nanoparticle) with uptake of nanoparticles through LDL receptors in endometriotic foci	[52]
Electrochemical immunosensor	Composite of Multiwalled carbon nanotube and magnetite nanoparticle	CA19-9	[42]
Flow cytometry	Natural Killer cells (NK)	NK cytotoxicity was determined through assay of 51 Cr release against K562 cells, and the expression of killer cell inhibitory receptors (KIRs, including NKB1, GL183, and EB6) in NK cells	[51,53]
Targeting microbiota	Microbiome of the gastrointestinal tract	Reduced microbiome diversity and an increased proportion of potentially pathogenic microbes using gene sequencing	[54]
Imaging	Iron oxide, polymeric, noble metal, silicate nanoparticles, hyaluronic acid (HA), and magnetic iron oxide nanoparticles	Labeled particles, dyes, endometriosis tissues	[43,55,56,57]
Magnetic resonance imaging	Nanoparticles with a peptide targeted to vascular endothelial growth factor receptor 2 (VEGFR-2)	Endometriotic tissue	[58]

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
