# Peer review of "A Comprehensive Review of Advanced Diagnostic Techniques for Endometriosis: New Approaches to Improving Women’s Well-Being"

_medicina, 2024, doi:10.3390/medicina60111866_

Round 1

Reviewer 1 Report

Comments and Suggestions for Authors

Dear Authors

The article entitled: Advances in endometriosis diagnostics techniques: the new approach to improve women’s well-being  

I find the manuscript very interesting and important however, I suggest some changes for its acceptance:   

Line 72, 73: Explain how polymorphism associated with age at menarche influences endometriosis, briefly.

Line 80- 84: You should describe acronyms: Erβ, Erα, TNFα, Greb-1, c-Myc

Overexpression of ERβ in the stromal EC inhibits the TNFα-mediated apoptosis, acts as a suppressor of ERα, induces interleukin-1, co-stimulates Ras-related estrogen1 regulated growth inhibitor and serum and glucocorticoid-regulated kinase 1 as key ERβ targets with co-stimulating prostaglandin E2 under the action of estradiol. De novo increase of E2 in endometriosis lesions affects the ratio of ERα and ERβ, impacting the inflammation and expression of some target genes such as Greb-1 and c-Myc that result in endometriosis progression

Line 97: Describe acronym: HLA

Line 107:  I suggest that you should write, which means the superficial peritoneal phenotype?

Line 139: Briefly describe why these markers are used for the identification of endometriosis: CA-199, interleukin-6, and urocortin.

In Table 2. Endometriosis diagnostic methods. In the row that mentions Targeting microbiota:  I suggest that explain this point better, i.e. how would endometriosis be diagnosed by microbiota composition? molecular biology tools? or nanotechnology tools?

Line 184: Describe NIR.

Line 292: Describe acronyms COG

Line 293: Describe acronyms GnRH

Conclusion: Explain the limitations of IA in conclusion versus medical assistance 

Author Response

Thank You for valuable comments on our manuscript. Changes are made according to your
suggestions.

Line 72, 73: Explain how polymorphism associated with age at menarche influences endometriosis,
briefly. Explanation is added.
Line 80- 84: You should describe acronyms: Erβ, Erα, TNFα, Greb-1, c-Myc. Overexpression of
ERβ in the stromal EC inhibits the TNFα-mediated apoptosis, acts as a suppressor of ERα, induces
interleukin-1, co-stimulates Ras-related estrogen1 regulated growth inhibitor and serum and
glucocorticoid-regulated kinase 1 as key ERβ targets with co-stimulating prostaglandin E2 under
the action of estradiol. De novo increase of E2 in endometriosis lesions affects the ratio of ERα and
ERβ, impacting the inflammation and expression of some target genes such as Greb-1 and c-Myc
that result in endometriosis progression. Acronyms described. 
Line 97: Describe acronym: HLA. Acronym described.
Line 107:  I suggest that you should write, which means the superficial peritoneal phenotype?
Information added.
Line 139: Briefly describe why these markers are used for the identification of endometriosis: CA-
199, interleukin-6, and urocortin. Information added.
In Table 2. Endometriosis diagnostic methods. In the row that mentions Targeting microbiota:  I
suggest that explain this point better, i.e. how would endometriosis be diagnosed by microbiota
composition? molecular biology tools? or nanotechnology tools? Corrections made.
Line 184: Describe NIR. Acronym described.
Line 292: Describe acronyms COG. Acronym described.
Line 293: Describe acronyms GnRH. Acronym described.
Conclusion: Explain the limitations of IA in conclusion versus medical assistance. Information
added.

Reviewer 2 Report

Comments and Suggestions for Authors

Dear Authors

Since Endometriosis is a common and lifelong gynecologic disease, your study is valuable.

1-please mention the type of study in the title.

2-Some of the references you cited were published more than five years ago. It is better to update them.

3-What are the inclusion and exclusion criteria for including references?

4-How did you determine the eligibility of studies included in your manuscript?

5-It is better for the conclusion to be more organized and summarized.

Author Response

Thank You for your insightful comments. Changes are made according to your suggestions.

1-please mention the type of study in the title.

Title changed.

2-Some of the references you cited were published more than five years ago. It is better to update
them.

References are updated.

3-What are the inclusion and exclusion criteria for including references?

In our manuscript, we established clear inclusion criteria to ensure the relevance and quality of the
studies referenced. We included studies published in reputable journals, research specifically
addressing diagnostic techniques for endometriosis, particularly those involving new technologies
and AI, DNA methylation or other epigenetic markers in endometriosis were included. We decided
to exclude include non-English studies, or studies which focused solely on treatment methods or
general overviews of endometriosis without a diagnostic focus.
To add, in addressing the inclusion and exclusion criteria for references in our manuscript, the focus
was on selecting studies and articles that presented significant advancements in nanotechnologies,
detection, treatment, endometriosis, sensors, AI. Specifically, we targeted innovations in diagnostic
and therapeutic methods from 2015 to 2025 (just a mathematical number) to capture the most recent

decade of advancements, emphasizing works that incorporated AI or other novel technologies.
Priority was given to studies with clinical trials and evidence-based findings to ensure clinical
applicability and reliability. We excluded and were trying to avoid conference work.

4-How did you determine the eligibility of studies included in your manuscript?

For the eligibility of the studies, we conducted a preliminary review of titles and abstracts to filter
out irrelevant studies based on our inclusion and exclusion criteria. For studies that passed the initial
screening, we performed a comprehensive full-text review to assess their relevance to the diagnostic
techniques for endometriosis. We then extracted data on diagnostic methodologies, technological
innovations, and findings related to AI and epigenetic regulation to ensure alignment with our
manuscript’s focus.
To determine eligibility, we assessed each study's relevance based on its methodological rigor, the
level of innovation, and the application of nanotechnologies or AI for endometriosis. Articles were
primarily sourced from trusted, peer-reviewed journals within Web of Science and other trusted
platforms to which our institution subscribes. This approach guaranteed access to high-quality
research that met our standards for relevance and evidence-based validity in endometriosis
diagnostics and treatment.

5-It is better for the conclusion to be more organized and summarized.

The conclusion was changed.

Reviewer 3 Report

Comments and Suggestions for Authors

Endometriosis remains a hot topic in gynecology and it impacts quality of life for a large number of women. Current advances in the field are of utmost interest. 

Still, the title only suggests diagnostic modalities yet the article comprises treatment courses.  The article fails to be anything else than a general and shallow review of the diverse studies in the field. It does not coagulate any pathways that the authors envision and it doesn't highlight traits that the readers should follow in the endeavour of optimal diagnosis and treatment. The conclusion paragraph is too brief to be able to  replace a discussion section and the disparte chapters within the article are not articulated in a unified vision.

Author Response

Endometriosis remains a hot topic in gynecology and it impacts quality of life for a large number of women. Current advances in the field are of utmost interest.
Still, the title only suggests diagnostic modalities yet the article comprises treatment courses.  The
article fails to be anything else than a general and shallow review of the diverse studies in the field.
It does not coagulate any pathways that the authors envision and it doesnt highlight traits that the
readers should follow in the endeavour of optimal diagnosis and treatment. The conclusion
paragraph is too brief to be able to replace a discussion section and the disparte chapters within the
article are not articulated in a unified vision.

Thank You for your opinion. Information is added in order to be clear in the main review message.